# Bond Durability of Two-Step HEMA-Free Universal Adhesive

**DOI:** 10.3390/jfb13030134

**Published:** 2022-08-29

**Authors:** Akimasa Tsujimoto, Nicholas G. Fischer, Wayne W. Barkmeier, Mark A. Latta

**Affiliations:** 1Department of Operative Dentistry, University of Iowa College of Dentistry, 801 Newton Rd., Iowa City, IA 52246, USA; 2Department of General Dentistry, Creighton University School of Dentistry, 2109 Cuming St., Omaha, NE 68102, USA; 3MDRCBB—Minnesota Dental Research Center for Biomaterials and Biomechanics, University of Minnesotas School of Dentistry, 16-212 Delaware St. SE, Minneapolis, MN 55455, USA

**Keywords:** dental bonding, dental debonding, dental restoration failure

## Abstract

The purpose of this study is to compare bond durability, in terms of fatigue bond strength, of a two-step HEMA-free universal adhesive and representative adhesives in each systematic category. The adhesives used in this study were OptiBond FL, Prime&Bond NT, Clearfil SE Bond 2, G2-Bond Universal, and Scotchbond Universal Plus Adhesive. Fatigue bond strength testing and scanning electron microscopy analysis of adhesively bonded enamel and dentin interfaces were performed. For the adhesives in etch-and-rinse mode, the enamel fatigue bond strength of the G2-Bond Universal adhesive was significantly higher than those of other adhesives, and the dentin fatigue bond strength of Prime&Bond NT was significantly lower than the others. For adhesives in self-etch mode, the enamel fatigue bond strengths of Clearfil SE Bond 2 and G2-Bond Universal were significantly higher than that of the Scotchbond Universal Plus Adhesive, and the dentin fatigue bond strength of G2-Bond Universal was significantly higher than Clearfil SE Bond 2 and the Scotchbond Universal Plus Adhesive. The two-step HEMA-free universal adhesive showed higher enamel and higher or equal dentin fatigue bond strength than other selected representative adhesive systems in etch-and-rinse mode and higher or equal enamel and higher dentin fatigue bond strength than adhesive systems in self-etch mode.

## 1. Introduction

Adhesive dentistry has made notable progress over recent decades, during which time the etch-and-rinse and self-etch approaches have both been established as standard, reliable methods [1]. Countless reviews and publications have compared and contrasted these systems, noting that simplified approaches, such as universal adhesive continue to gain popularity [2], but that multiple-bottle systems remain some of the best performing adhesives ever developed. Adhesives combining all properties of the etch-and-rinse and self-etch modes have been envisioned, such as universal adhesives with alternative techniques or additional strategies [3], but still do not present the adhesive which is clearly better in bonding performance than existing adhesive systems.

Among the etch-and-rinse adhesive systems, OptiBond FL (Kerr, Orange, CA, USA), which is a three-step etch-and-rinse adhesive system and is recognized as one of the gold standards of multiple-bottle system, has been recognized as the gold standard [4]. This gold standard etch-and-rinse adhesive includes a highly hydrophobic adhesive which contains glycerol phosphate dimethacrylate (GPDM). This monomer also includes self-etch and universal adhesive [5] and can chemically react with hydroxyapatite in the etched and primed enamel and dentin surface to increase bond durability [6]. OptiBond FL was launched in 1995 and was reported as the number one adhesive system in a benchmark paper for adhesive dentistry [4]. Despite significant developments in the field and the release of novel adhesive systems in the 25 years since OptiBond FL was released, no new adhesive has offered a clearly superior bond performance, and thus, it is still the adhesive of choice based on personal preference [7].

Among the self-etch adhesive systems that do not require phosphoric acid etching, Clearfil SE Bond (Kuraray Noritake Dental, Tokyo, Japan) released in 1991 and is considered the gold standard in this category [8]. The primers and adhesives of these systems typically include 10-methacryloyloxydecyl dihydrogen phosphate (10-MDP) which creates a stable and strong bond formed by nanolayers with substrates’ calcium [9]. The primers and adhesives of these systems typically include 10-MDP which creates a strong and stable chemical bond with substrates’ calcium forming nanolayers of 10-MDP-Ca [5].

A more recent approach to bonding, which has gained significant popularity over the last ten years, is the universal adhesive system, a simplified adhesive system that works with or without phosphoric acid etching [10]. After Kuraray’s patent on 10-MDP expired, 3M Oral Care launched Scotchbond Universal Adhesive in 2013 and since then many manufacturers have followed and developed this category of adhesive system. 

Many of these adhesives, including those mentioned above, have relied on 2-hydroxyethyl methacrylate (HEMA) to enhance bonding with dentin [11]. HEMA is hydrophilic, highly compatible with dentin, and is also capable of easily penetrating demineralized substrate. On the other hand, its hydrophilicity makes it susceptible to hydrolysis and sorption, and it is also known to give rise to allergic reactions [12]. As a result, manufacturers have recently started to introduce HEMA-free adhesives.

These types of systems all have strengths and weaknesses. Generally, adhesives in etch-and-rinse mode are hydrophobic and thus durable adhesives, but must be used with phosphoric acid etching and are necessary for enamel bonding, which is not always the most appropriate approach and is recognized as aggressive for dentin bonding [4]. The self-etch systems avoid the phosphoric acid etching to dentin and can be used with the self-etch or selective-etch approaches, but they are more hydrophilic, and thus are prone to degradation [7]. Finally, the universal adhesives can be used with or without the phosphoric acid etching and have more hydrophilicity than multi-step adhesives [13].

If it were possible to combine all of these positive features in a HEMA-free adhesive, the resulting adhesive might constitute a significant advance in bonding technology. A two-step system using a primer based on universal adhesives could be used with or without phosphoric acid etching, as appropriate, which would make it more flexible [14]. In addition, the adhesive itself could be made hydrophobic, which should increase the durability [15]. The exclusion of HEMA from both the primer and adhesive should reinforce this durability, while reducing allergenicity. A new type of adhesive taking this approach, the two-step HEMA-free universal adhesive, G2-Bond Universal from GC (Tokyo, Japan), has recently been released.

Although previous studies have investigated some aspects of the performance of this new adhesive type, such as marginal adaptation and initial bond strength [16,17,18], there is no report of fatigue bond strength of G2-Bond Universal being compared to representative adhesives. This study considers in vitro bonding of a resin composite (Filtek Supreme Ultra, 3M Oral Care) to dentin and enamel (the Problem), using the two-step HEMA-free universal adhesive (the Intervention), and comparing it to representative existing adhesives of the other types of adhesives (the Comparators) in terms of fatigue bond strength (the Outcome). Thus, the PICO question is whether the two-step HEMA-free universal adhesive will show a significantly higher fatigue bond strength than any or all of the existing representative adhesive systems.

## 2. Materials and Methods

Adhesives used are shown in Table 1. A three-step etch-and-rinse adhesive, OptiBond FL (Kerr, Orange, CA, USA), a two-step E&R adhesive, Prime&Bond NT (Dentsply Sirona, Charlotte, NC, USA), a two-step self-etch adhesive, Clearfil SE Bond 2, a two-step universal adhesive, G2-BOND Universal (GC), and a one-step universal adhesive, Scotchbond Universal Plus (3M Oral Care, St. Paul, MN, USA) were used to bond a resin composite (Filtek Supreme Ultra, 3M Oral Care) to both enamel and dentin. In addition, phosphoric acid etchant (UltraEtch, Ultradent Products, South Jordan, UT, USA) was used. Adhesives, except for the universal adhesives, were applied in either etch-and-rinse or self-etch mode and universal adhesives were used in both modes based on the manufacturers’ instructions in this laboratory study. 

### 2.1. Shear Fatigue Bond Strength Testing

Extracted, non-carious third molars from humans were used as substrates. The use of human molars was approved by the Institutional Review Board (IRB) of the University of Iowa (IRB ID#: 20220313-3 March 2022) and Creighton University (IRB ID#: 760765-1-22 May 2015). Our universities waived the requirement for ethical approval and written informed consent for participants in this study due to the used samples being de-identified, anonymized, and discarded from use for patient care only. This study was conducted in accordance with the Declaration of Helsinki of 1975, revised in 2013.

Enamel and dentin surfaces were prepared by separating the molars mesially–distally after removing approximately 2/3rd of the root. The separated facial and lingual molar sections were placed in 25 mm in diameter and height phenolic rings with acrylic resin (Fastry Custom Tray and Acrylic Base Plate Material, Keystone Industries, Gibbstown, NJ, USA). Teeth were placed in the resin to keep the dentin tubules vertical. The center of the facial and lingual tooth surfaces was flattened and polished up to 4000 grit using silicon carbide papers under water coolant to reach shallow, and standardized, enamel or dentin. The motivation to use this highly polished surface is to minimize the influence of surface roughness of base surface. 

Twenty specimens were made for each group for the shear fatigue bond strength test. Number of specimens for the shear fatigue bond strength was followed by the discussion in the review paper of the testing [1]. Stainless-steel rings machined with dimensions of 2.38 (inner diameter), 4.70 mm (outer diameter), and 2.62 mm (edge thickness) were fixed on the surfaces treated with the different adhesives in either etch-and-rinse or self-etch mode, according to the manufacturers’ instructions. The resin composite was placed in stainless-steel rings using a custom holder. An LED light curing unit (Valo Cordless, Ultradent Products; 40 s) was used to photopolymerize. Specimens were then stored for 24 h at 37 °C in distilled water before testing.

A dynamic fatigue testing system (ElectroPuls E1000, Instron, Norwood, MA, USA) was used to load the enamel and dentin-bonded specimens based on our previously established method [6,19]. The initial load force, using the staircase test method for fatigue bond strength testing, was programmed at ca. 50% of the ultimate strength measured in initial shear bond strength testing. A chisel-shaped metal rod applied the cyclical force, which was applied as a sine wave for 50,000 cycles or until failure occurred at 20 Hz with a lower limit at approximately 0 [19]. The load was incrementally changed after each sample: either increased by circa 10% for specimens that survived or decreased by circa 10% for specimens that failed. The loading force that produced 50% failures was calculated, normalized to the surface area, based on Draughn [20], and is referred to as shear fatigue bond strength based on the bonding surface area of 4.37 mm^2^. Specimen geometry and fatigue testing are summarized in Figure 1.

### 2.2. Shear Bond Strength of Survivors Testing

After completion of fatigue testing as described above, the shear bond strength of the specimens that survived testing was determined. Specimens were monotonically loaded until failure at 1 mm min^−1^, based on our previously established method [1,10,21].

### 2.3. SEM Observations

Adhesive interfaces of the adhesive systems were visualized using field-emission scanning electron microscopy (ERA 8800FE, Elionix, Tokyo, Japan). Bonded specimens were split perpendicularly near the center of the specimen in half. The sectioned specimens were embedded in resin epoxy (Epon 812, Nisshin EM, Tokyo, Japan) in a paper mouthwash cup. The embedded specimens were adjusted to be 6 × 6 mm square and 3 mm in height. The adhesive interfaces were mirror-polished to 0.25 µm diamond paste (DP-Paste, Struers, Copenhagen, Denmark). Specimens were dehydrated in tert-butyl alcohol after ultrasonic cleaning for 30 s and then freeze-dried. The interface surfaces of dried specimens were treated with Ar ion beam etching (EIS-200 ER, Elionix) perpendicular to the surface to make the material differences clearer. The etched interfaces were coated with gold film using a coating machine (Quick Coater Type SC-701, Sanyu Electron, Tokyo, Japan) and visualized using SEM with an accelerating voltage of 10 kV.

### 2.4. Statistical Analysis

Normality was validated using a Shapiro–Wilk test on the shear bond strength values. As normality was confirmed, the results were analyzed by analysis of variance (ANOVA) paired with a Tukey’s post hoc test. A modified *t*-test (pooled variance) with Bonferroni correction was utilized to compare the fatigue bond strengths, given ANOVA is inappropriate for these data. Data are shown as mean ± standard deviation.

## 3. Results

### 3.1. Enamel and Dentin Fatigue Bond Strength of Adhesives in Etch-and-Rinse Mode

The experimental set-up is shown in Figure 1. Fatigue bond strength test results using the staircase method for etch-and-rinse mode to enamel and dentin are shown in Figure 2 and Figure 3. The incrementally adjusted (circa 10%) loading force can be seen. The fatigue bond strengths of all the tested adhesive systems in etch-and-rinse mode are shown in Table 2. The fatigue bond strengths were 17.8 MPa to enamel and 20.4 MPa to dentin for OptiBond FL, 21.0 MPa to enamel and 11.4 MPa to dentin for Prime&Bond NT, 24.6 MPa to enamel and 20.7 MPa to dentin for G2-Bond Universal (etch-and-rinse mode), and 19.7 MPa to enamel and 17.5 MPa to dentin for Scotchbond Universal Plus (etch-and-rinse mode). 

Among the adhesives in etch-and-rinse mode, the enamel fatigue bond strength of OptiBond FL was significantly less than Prime&Bond NT, G2-Bond Universal (etch-and-rinse mode), and the Scotchbond Universal Plus Adhesive (etch-and-rinse mode). In addition, the enamel fatigue bond strength of the G2-Bond Universal adhesive (etch-and-rinse mode) was significantly greater than those of other adhesives in etch-and-rinse mode. On the other hand, the dentin fatigue bond strength of Prime&Bond NT was significantly less than other adhesives in etch-and-rinse mode. Shear bond strengths of survivors of adhesives in etch-and-rinse mode to enamel showed no statistically significant differences. Shear bond strength of survivors to dentin of Prime&Bond NT was significantly lower than other adhesives in etch-and-rinse mode. The percentages of survivor specimens were 45–65% and fatigue bond strength/shear bond strength of survivors were 40–55%.

Among the fatigue bond strength and shear bond strength of survivors to enamel and dentin, there is no difference for OptiBond FL, G2-Bond Universal (etch-and-rinse mode), and the Scotchbond Universal Plus Adhesive (etch-and-rinse mode), unlike for Prime&Bond NT.

### 3.2. SEM Observations of Dentin-Adhesive Interface of Etch-and-Rinse Adhesives

SEM visualization of the dentin-adhesive interfaces of the adhesives in etch-and rinse mode are presented in Figure 4, Figure 5, Figure 6 and Figure 7 and revealed excellent adaptation across all adhesives. Adhesive layer thickness was different depending on adhesive; adhesive layer thicknesses of OptiBond FL and G2-Bond Universal were around 50 µm thick; that of Prime&Bond NT was around 30 µm thick and that of Scotchbond Universal Plus Adhesive was around 10 µm thick. On the other hand, the hybrid layer thickness of all in etch-and-rinse mode was approximately 2–3 µm thick.

### 3.3. Enamel and Dentin Fatigue Bond Strength of Adhesives in Self-Etch Mode

Fatigue bond strength test results for adhesives in self-etch mode to enamel and dentin, determined from the staircase method, are shown in Figure 8 and Figure 9. The fatigue bond strengths of all the tested adhesive systems in self-etch mode are shown in Table 3. The fatigue bond strengths were 21.5 MPa to enamel and 23.2 MPa to dentin for Clearfil SE Bond 2, 21.3 MPa to enamel and 27.2 MPa to dentin for G2-Bond Universal (self-etch mode), and 12.1 MPa to enamel and 13.7 MPa to dentin for Scotchbond Universal Plus Adhesive (self-etch mode). 

Looking at the enamel fatigue bond strength of adhesives in self-etch mode, the values for Clearfil SE Bond 2 and G2-Bond Universal (self-etch mode) were significantly greater than the Scotchbond Universal Plus Adhesive (self-etch mode). Among the adhesives applied in self-etch mode, the fatigue bond strength of G2-Bond Universal in self-etch mode to dentin was significantly greater than Scotchbond Universal Plus (self-etch mode) and Clearfil SE Bond 2. In addition, the fatigue bond strength to dentin of Clearfil SE Bond 2 was significantly greater than the Scotchbond Universal Plus Adhesive (self-etch mode).

Among the fatigue bond strength to enamel and dentin, G2-Bond Universal (self-etch mode) showed significantly greater fatigue bond strength to dentin than that to enamel. There is no difference in fatigue bond strength to dentin and enamel in Clearfil SE Bond 2 and the Scotchbond Universal Plus Adhesive (self-etch mode).

Shear bond strengths of survivors adhesively bonded to enamel and dentin of both Clearfil SE Bond 2 and G2-Bond Universal (self-etch mode) were significantly greater than that of Scotchbond Universal (self-etch mode). Percentages of survivor specimens were 40–65% and fatigue bond strengths/shear bond strengths of survivors were 43–60%.

Among the shear bond strengths of survivors to enamel and dentin, all three tested adhesives in self-etch mode showed significantly greater shear bond strength to dentin than that to enamel.

### 3.4. SEM Observations of Dentin-Adhesive Interface of Self-Etch Adhesives

SEM visualization of the dentin-adhesive interfaces of adhesives in self-etch mode are presented in Figure 10, Figure 11 and Figure 12 and revealed excellent adaptation across all adhesives. The adhesive layer thicknesses of Clearfil SE Bond 2 and G2-Bond Universal (self-etch mode) were similar and around 50 µm thick. On the other hand, the adhesive layer thickness of the Scotchbond Universal Plus Adhesive (self-etch mode) was around 10 µm thick and thinner than that of G2-Bond Universal and Clearfil SE Bond 2 (self-etch mode). Hybrid layers for three tested self-etch adhesives were not clearly seen in the SEM visualization of the dentin-adhesive interfaces. 

## 4. Discussion

The bond durability, in terms of fatigue bond strength, of G2-Bond Universal bonded to both enamel and dentin was consistently equal to or better than that of other adhesive systems, regardless of the etching mode. In this study, representative adhesives were selected from each type of adhesive system for comparison with the bond durability of G2-Bond Universal. 

OptiBond FL, Prime&Bond NT, and the Scotchbond Universal Plus Adhesive (etch-and-rinse mode) were selected as adhesives in the etch-and-rinse mode for comparison. The key difference among these systems is the hydrophilicity of the adhesive agents; OptiBond FL and G2-Bond Universal are more hydrophobic than Prime&Bond NT and the Scotchbond Universal Plus Adhesive, due to the lack of water in the composition. G2-Bond Universal is specifically designed to be more hydrophobic than OptiBond FL due to the lack of HEMA. The enamel fatigue bond strength of G2-Bond Universal (etch-and-rinse mode) was significantly higher than those of OptiBond FL, Prime&Bond NT, and the Scotchbond Universal Plus Adhesive (etch-and-rinse mode), while the values for Prime&Bond NT and Scotchbond Universal Plus were significantly greater than that of OptiBond FL. These results suggest the increased hydrophobicity of G2-Bond does not impede adhesive performance.

In the case of etched enamel bonding, the morphological and interfacial characteristics of enamel are changed by phosphoric acid etching [22]. It has been thought that the most important contribution to bond durability between adhesive and etched enamel is from micro-mechanical interlocking due to the penetration and polymerization of adhesive agents within the honeycomb microstructure of the etched surface [23]. Thus, it is thought that bond strength arises primarily from mechanical interlocking regardless of the type of the adhesive; this has been known since 1955 [22]. However, it is important to consider all aspects of adhesion if further improvements in adhesive dentistry are to be achieved. Indeed, compatible hydrophilicity between the adhesive agent and the primed or non-primed etched enamel surface may be important [23].

In the cases of Prime&Bond NT and the Scotchbond Universal Plus Adhesive, the adhesives are designed for a more hydrophilic surface due to the lack of primer. When Prime&Bond NT and the Scotchbond Universal Plus Adhesive are directly applied to the etched enamel surface, that surface is highly hydrophilic due to the exposure of hydroxyl groups. Therefore, the compatibility between the adhesive agents and etched enamel is high, and the adhesive agent can directly penetrate into the etched surface without primer application to create a stable bonding interface.

On the other hand, the adhesive agents of OptiBond FL and G2-Bond Universal, which use a primer, are designed to be more hydrophobic to secure bond durability over time. The difference between OptiBond FL and G2-Bond Universal is the hydrophilicity of the primer. Although the main contents of both primers are fundamentally similar—acidic functional monomers, water, fillers, solvents, and photoinitiators—the important differences in primer composition between OptiBond FL and G2-Bond Universal are the presence or absence of HEMA, concentrations of contents, and hydrophilicity. Therefore, the hydrophilicity of the primer and adhesive itself is higher in OptiBond FL than in G2-Bond Universal. The susceptibility of HEMA to bond degradation is well-documented [11]. The bonding of G2-Bond Universal to etched enamel is based on changing the surface characteristics of the etched enamel to be hydrophobic and establishing a more hydrophobic adhesive layer in the adhesive interface. The results suggest this thorough creation of a hydrophobic surface was more effective in securing fatigue strength to etched enamel than the less hydrophobic environment created by OptiBond FL.

Considering the dentin fatigue bond strength of adhesives in etch-and-rinse mode, G2-Bond Universal (etch-and-rinse mode) showed a similar dentin fatigue bond strength to OptiBond FL and the Scotchbond Universal Plus Adhesive (etch-and-rinse mode) and a significantly higher dentin fatigue bond strength than Prime&Bond NT. When bonding to etched dentin, the most important bonding mechanism is the establishment of a hybrid layer with demineralized dentin to reinforce the adhesive layer [24]. The results suggest that the hydrophobic adhesives were capable of penetrating between the exposed collagen fibers and reinforcing the dentin surface to create a stronger bond. On the other hand, the role that thickness of the adhesive layers plays on the dentin fatigue bond strength of the hydrophilic adhesives appears to be large. The adhesive layer in the Scotchbond Universal Plus Adhesive was generally less than 10 µm thick, while that of Prime&Bond NT was three times thicker and 30 µm thick. The thicker adhesive layer may well be responsible for the much lower dentin fatigue strength of Prime&Bond NT.

Clearfil SE Bond 2 and the Scotchbond Universal Plus Adhesive (self-etch mode) were selected for comparison to the bond durability of G2-Bond Universal in self-etch mode. For enamel bonding in self-etch mode, the acidity of the adhesive itself is important to secure basic mechanical interdigitation to enamel. The pH of the primer is 2.0 for Clearfil SE Bond 2 and 1.5 for G2-Bond Universal, and the pH of adhesive is 2.7 for the Scotchbond Universal Plus Adhesive. Unlike etched enamel, ground enamel itself is hydrophobic, thus the compatibility between the adhesive agents and ground enamel is higher in Clearfil SE 2 and G2-Bond Universal, in addition to the stronger acidity of the adhesive systems. These two factors may explain why Clearfil SE Bond 2 and G2-Bond Universal (self-etch mode) showed higher enamel fatigue strength than the Scotchbond Universal Plus Adhesive (self-etch mode).

That said, for dentin fatigue bond strength values, G2-Bond Universal (self-etch mode) showed significantly higher values than Clearfil SE Bond 2 and the Scotchbond Universal Plus Adhesive (self-etch mode). The 2–3-micrometer hybrid layer that is found in etch-and-rinse mode is not present in self-etch mode, and so possible influence of the reinforcement of the demineralized dentin is limited. Thus, the creation of a stronger adhesive layer becomes important and the effects of this are clear. As mentioned earlier, G2-Bond Universal is more hydrophobic than Clearfil SE Bond 2 and much more hydrophobic than Scotchbond Universal Plus. The lack of a measurable difference in fatigue bond strength between Clearfil SE Bond 2 and the Scotchbond Universal Plus Adhesive may be due to the weakness of Scotchbond Universal Plus being a higher hydrophilicity of the adhesive layer. In contrast, the higher fatigue bond strength of G2-Bond Universal (self-etch mode) may be ascribed to its more hydrophobic character and the gradually increasing strength of each of the layers of dentin, primer, adhesive, and resin-based composite. The HEMA-free primer and adhesive are critical to both factors, with the absence of HEMA increasing both the hydrophobicity of the primer and adhesive and the strength of the cured layers. 

Considering all the results, G2-Bond Universal showed the highest fatigue bond strength to etched enamel (at 24.6 MPa) and to ground dentin (at 27.3 MPa). Although there is limitation in this study, such as the methodology because the testing was not able to fully simulate aging phenomena occurring in the oral cavity in short periods of time and only evaluated the fatigue bond strength with G2-Bond Universal and representative adhesives, the use of G2-Bond Universal in the selective etching mode may be the best way to secure high fatigue bond strength compared to other representative adhesives. However, one limitation is the dentin tubule orientation relative to the applied load. Some studies have shown that shear bond strength is dependent on tubule orientation [25,26]. Here, we attempted to keep the dentin tubules vertical, but future research should evaluate the influence of dentin tubule orientation on G2-Bond Universal bond durability.

## 5. Conclusions

Results from this study indicated that the two-step HEMA-free universal adhesive, G2-Bond Universal, showed higher enamel and higher or equal dentin fatigue bond strength than other representative adhesives in etch-and-rinse mode and higher or equal enamel and higher dentin fatigue bond strength than adhesive systems in self-etch mode.

## Figures and Tables

**Figure 1 jfb-13-00134-f001:**
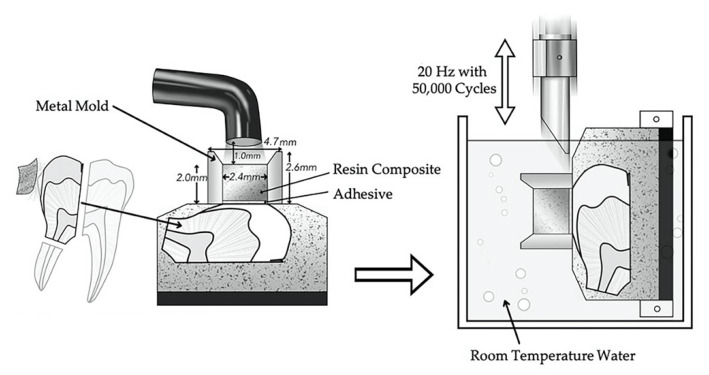
Schematic illustration of experimental setup for shear fatigue bond strength. This schematic is modified from open access [19], original © Operative Dentistry, Inc.

**Figure 2 jfb-13-00134-f002:**
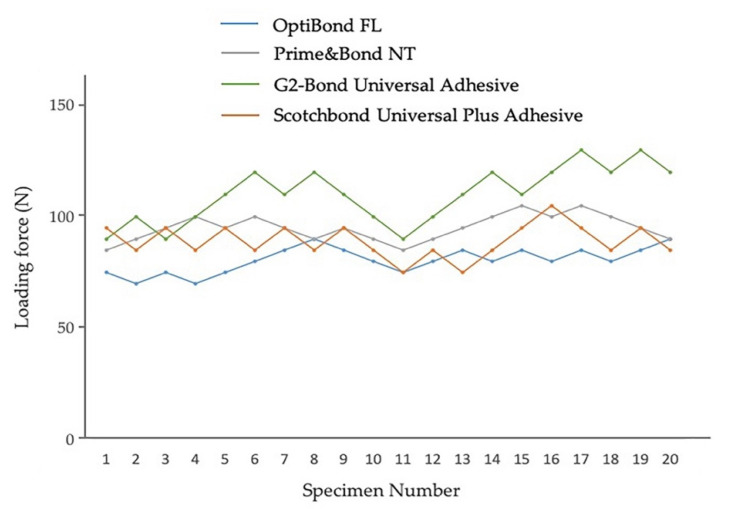
Enamel fatigue bond strength loading force results using the staircase method for adhesives in etch-and-rinse mode. The loading forces for 1st specimens were determined by the results of shear bond strength testing [OptiBond FL: 33 MPa; Prime&Bond NT: 38 MPa; G2-Bond Universal (etch-and-rinse mode): 40 MPa; Scotchbond Universal Plus Adhesive (etch-and-rinse mode): 42 MPa]. The average values were the numbers truncating the numbers beyond the first decimal point.

**Figure 3 jfb-13-00134-f003:**
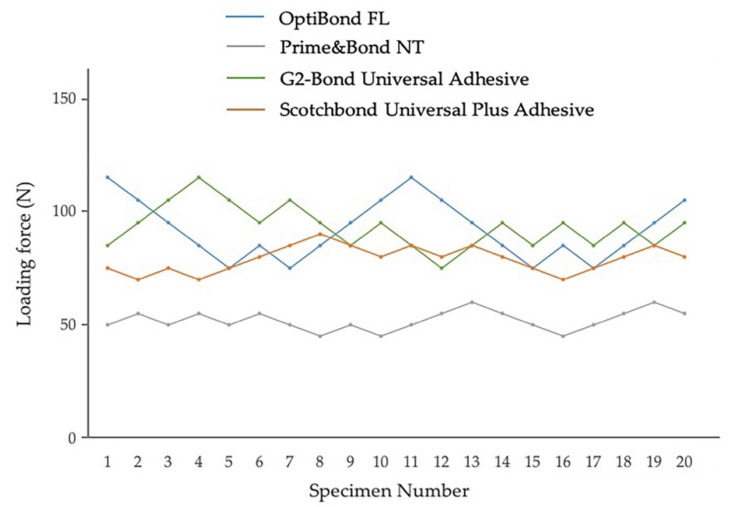
Dentin fatigue bond strength loading force results using the staircase method for adhesives in etch-and-rinse mode. The loading forces for 1st specimens were determined by the results of average shear bond strength value [OptiBond FL: 51 MPa; Prime&Bond NT: 22 MPa; G2-Bond Universal (etch-and-rinse mode): 40 MPa; Scotchbond Universal Plus Adhesive (etch-and-rinse mode): 42 MPa]. The average values were the numbers truncating the numbers beyond the first decimal point.

**Figure 4 jfb-13-00134-f004:**
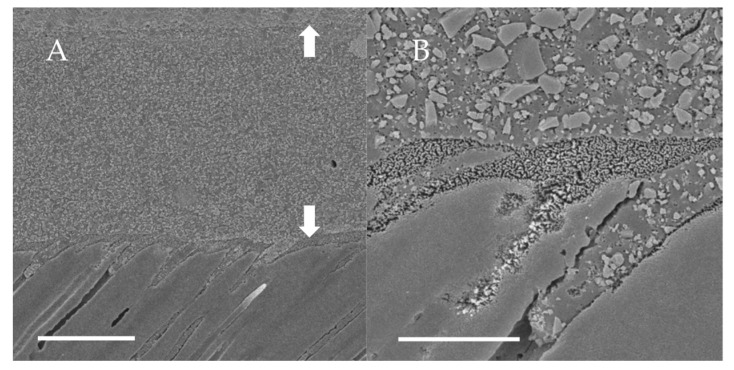
SEM observation of adhesive interface for OptiBond FL to dentin. (**A**): ×1000, scale indicates 20 µm. Arrows indicate the adhesive layer; (**B**): ×5000, scale indicates 5 µm.

**Figure 5 jfb-13-00134-f005:**
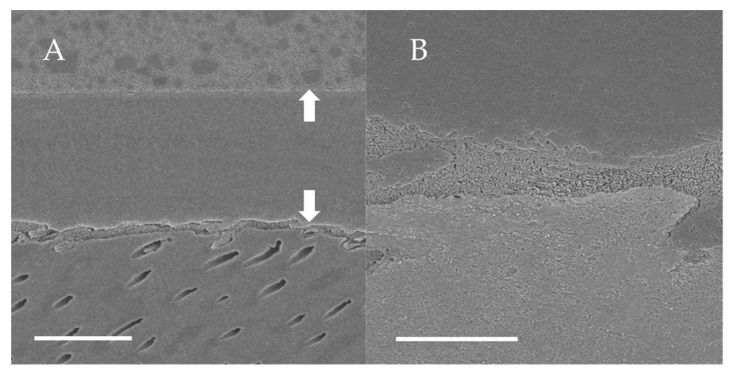
SEM observation of adhesive interface for Prime&Bond NT to dentin. (**A**): ×1000, scale indicates 20 µm. Arrows indicate the adhesive layer; (**B**): ×5000, scale indicates 5 µm.

**Figure 6 jfb-13-00134-f006:**
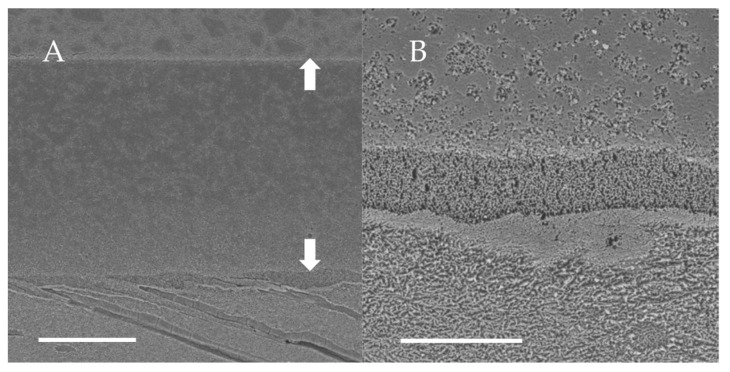
SEM observation of adhesive interface for G2-Bond Universal to dentin. (**A**): ×1000, scale indicates 20 µm. Arrows indicate the adhesive layer; (**B**): ×5000, scale indicates 5 µm.

**Figure 7 jfb-13-00134-f007:**
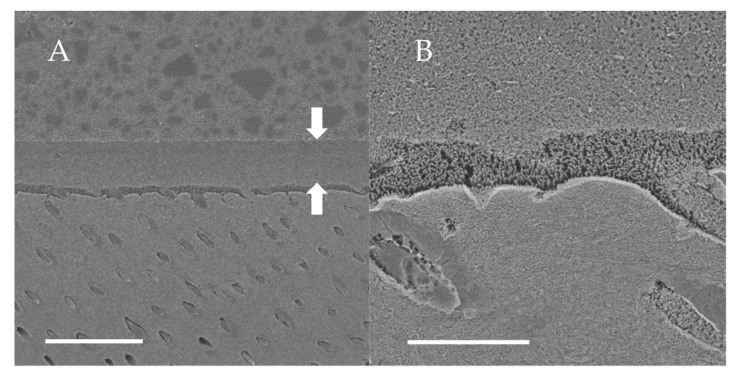
SEM observation of adhesive interface for Scotchbond Universal Plus to dentin. (**A**): ×1000, scale indicates 20 µm. Arrows indicate the adhesive layer; (**B**): ×5000, scale indicates 5 µm.

**Figure 8 jfb-13-00134-f008:**
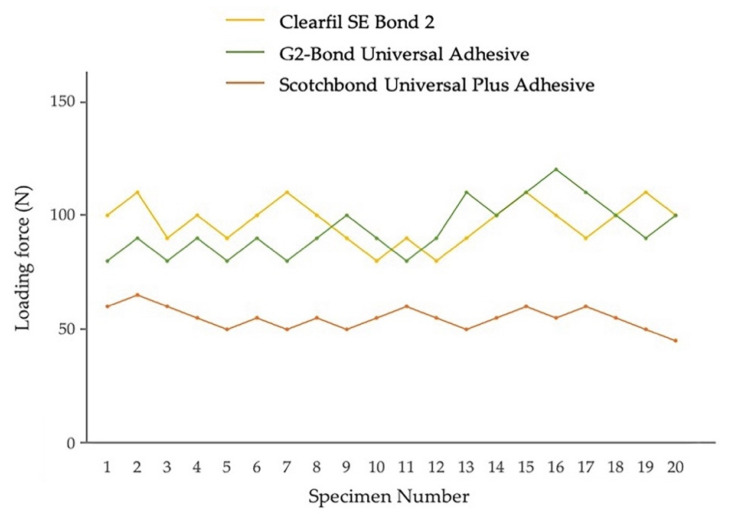
Enamel fatigue bond strength loading force results using the staircase method for adhesives in self-etch mode. The loading forces for 1st specimens were determined by the results of average shear bond strength value [Clearfil SE Bond 2: 44 MPa; G2-Bond Universal (self-etch mode): 35 MPa; Scotchbond Universal Plus Adhesive (self-etch mode): 27 MPa]. The average values were the numbers truncating the numbers beyond the first decimal point.

**Figure 9 jfb-13-00134-f009:**
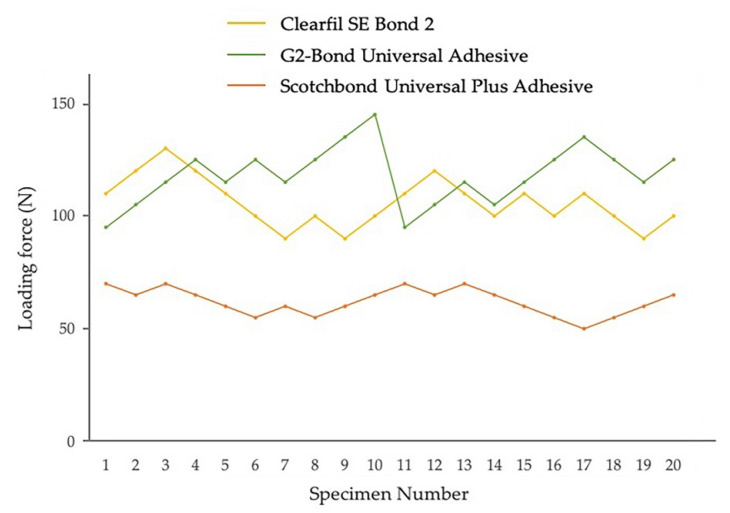
Dentin fatigue bond strength loading force results using the staircase method for adhesives in self-etch mode. The loading forces for 1st specimens were determined by the results of average shear bond strength value [Clearfil SE Bond 2: 49 MPa; G2-Bond Universal (self-etch mode): 43 MPa; Scotchbond Universal Plus Adhesive (self-etch mode): 31 MPa]. The average values were the numbers truncating the numbers beyond the first decimal point.

**Figure 10 jfb-13-00134-f010:**
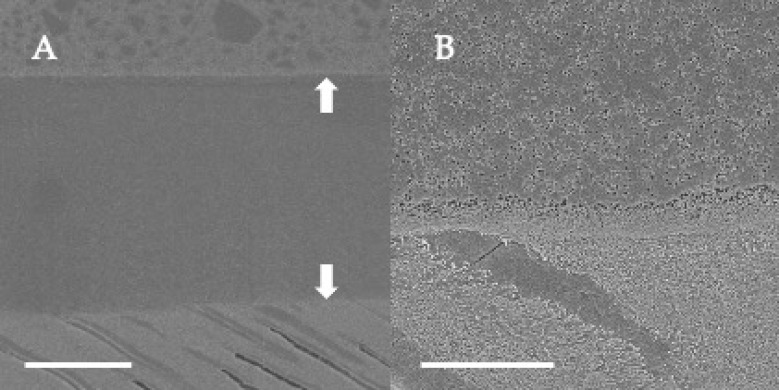
SEM observation of adhesive interface for Clearfil SE Bond 2 to dentin. (**A**) ×1000, scale indicates 20 µm. Arrows indicate the adhesive layer; (**B**) ×5000, scale indicates 5 µm.

**Figure 11 jfb-13-00134-f011:**
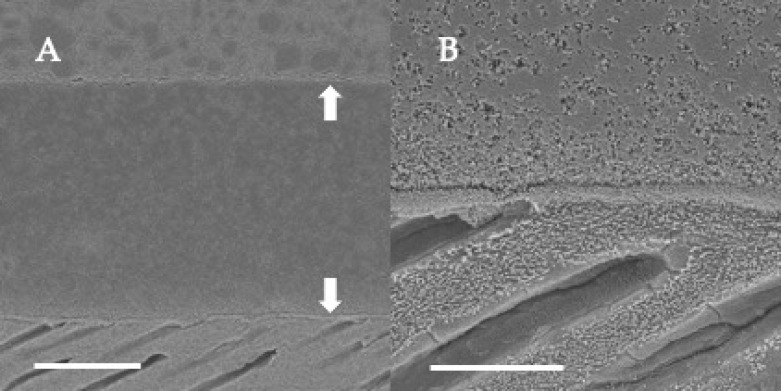
SEM observation of adhesive interface for G2-Bond Universal to dentin. (**A**) ×1000, scale indicates 20 µm. Arrows indicate the adhesive layer; (**B**) ×5000, scale indicates 5 µm.

**Figure 12 jfb-13-00134-f012:**
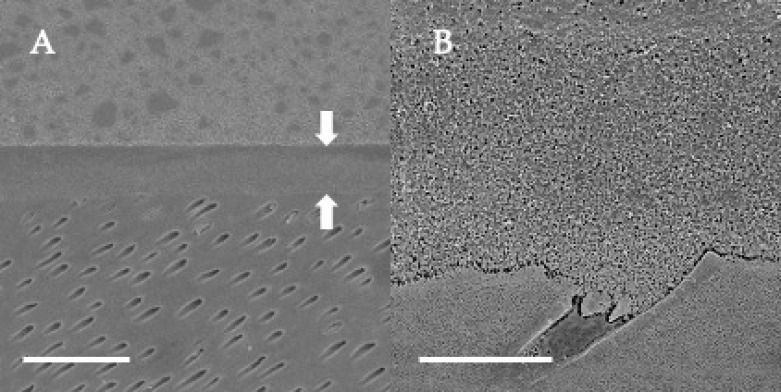
SEM observation of adhesive interface for Scotchbond Universal Plus Adhesive to dentin. (**A**) ×1000, scale indicates 20 µm. Arrows indicate the adhesive layer; (**B**) ×5000, scale indicates 5 µm.

**Table 1 jfb-13-00134-t001:** Adhesives used in this study.

Adhesive System	Type of Adhesive System	Main Components
OptiBond FL	Three-step Etch-and-rinse Adhesive	Primer: GPDM, 2-HEMA, ethanol water, initiators
		Adhesive: Bis-GMA, UDMA, TEGDMA, GDMA, 2-HEMA, initiators, fillers
Prime&Bond NT	Two-step Etch-and-rinse Adhesive	Adhesive: PENTA, UDMA, cetylamine hydrofluoride, acetone, fillers, initiators
Clearfil SE Bond 2	Two-step Self-etch Adhesive	Primer: 10-MDP, HEMA, water, initiators
		Adhesive: 10-MDP, 2-HEMA, Bis-GMA, initiators, fillers
G2-Bond Universal	Two-step Universal Adhesive	Primer: 4-MET, 10-MDP, 10-MDTP, dimethacrylate monomer, acetone, water, initiators, fillers
		Adhesive: dimethacrylate monomer, Bis-GMA, filler, photoinitiator
Scotchbond Universal Plus Adhesive	One-step Universal Adhesive	Adhesive: Bis-GMA, 10-MDP, 2-HEMA, Vitrebond copolymer, ethanol, water, initiators, fillers

Bis-GMA: bisphenol A glycidyl methacrylate; GDMA: glycerol phosphate dimethacrylate; GPDM: glycerol phosphate dimethacrylate; PENTA: penta-acrylate ester; TEGDMA: triethyleneglycol dimethacrylate; UDMA: urethane dimethacrylate; 2-HEMA: 2-hydroxyethyl methacrylate; 4-MET: 4-Methacryloyloxyethyl trimellitate; 10-MDP: 10-methacryloyloxydecyl dihydrogen phosphate; 10-MDTP; 10-methacryloyloxydecyl dihydrogen thiophosphate.

**Table 2 jfb-13-00134-t002:** Fatigue bond strength (FBS) and shear bond strength of survivors (SBSS), ratio of failure specimens, and FBS/SBSS to enamel and dentin of etch-and-rinse adhesive and universal adhesives in etch-and-rinse mode.

Adhesive System	Type of Adhesive System	Fatigue Bond Strength (FBS)		Shear Bond Strength of Survivors (SBSS)		Ratio ofSurvivor Specimens		Ratio of FBS/SBSS	
		Enamel	Dentin	Enamel	Dentin	Enamel	Dentin	Enamel	Dentin
OptiBond FL	Three-step Etch-and-rinse Adhesive	17.8 (2.0) ^a,A^	20.4 (5.0) ^a,A^	40.9 (9.4) ^a,A^	44.4 (8.9) ^a,A^	0.65	0.50	0.44	0.46
Prime&Bond NT	Two-step Etch-and-rinse Adhesive	21.0 (1.9) ^b,A^	11.4 (1.5) ^b,B^	41.8 (9.3) ^a,A^	25.1 (4.2) ^b,B^	0.50	0.45	0.50	0.45
G2-Bond Universal	Two-step Universal Adhesive	24.6 (4.5) ^c,A^	20.7 (2.5) ^a,A^	44.4 (8.7) ^a,A^	45.7 (10.0) ^a,A^	0.50	0.55	0.55	0.45
Scotchbond Universal Plus Adhesive	One-step Universal Adhesive	19.7 (1.1) ^b,A^	17.5 (2.0) ^c,A^	41.6 (3.9) ^a,A^	43.6 (7.5) ^a,A^	0.50	0.55	0.47	0.40

Standard deviation of fatigue bond strength values indicated in round brackets. The same lowercase letters in the individual column showed no statistically significant difference (*p* > 0.05). The same capital letters in the individual row showed no statistically significant difference (*p* > 0.05).

**Table 3 jfb-13-00134-t003:** Fatigue bond strength (FBS) and shear bond strength of survivors (SBSS), ratio of failure specimens, and FBS/SBSS to enamel and dentin of etch-and-rinse adhesive and universal adhesives in self-etch mode.

Adhesive System	Type of Adhesive System	Fatigue Bond Strength (FBS)		Shear Bond Strength of Survivors (SBSS)		Ratio of Failure Specimens/Survivors		Ratio of FBS/SBSS	
		Enamel	Dentin	Enamel	Dentin	Enamel	Dentin	Enamel	Dentin
Clearfil SE Bond2	Two-step Etch-and-rinse Adhesive	21.5 (2.3) ^a,A^	23.2 (3.4) ^a,A^	40.3 (5.2) ^a,A^	53.8 (8.6) ^a,B^	0.55	0.45	0.53	0.43
G2-Bond Universal	Two-step Universal Adhesive	21.3 (3.6) ^a,A^	27.2 (2.9) ^b,B^	36.4 (5.8) ^a,A^	45.7 (10.0) ^a,B^	0.55	0.65	0.59	0.60
Scotchbond Universal Plus Adhesive	One-step Universal Adhesive	12.1 (1.0) ^b,A^	13.7 (1.7) ^c,A^	28.4 (3.7) ^b,A^	34.6 (6.5) ^b,B^	0.40	0.40	0.50	0.40

Standard deviation of fatigue bond strength values indicated in round brackets. The same lowercase letters in the individual column showed no statistically significant difference (*p* > 0.05). The same capital letters in the individual row showed no statistically significant difference (*p* > 0.05).

## Data Availability

Not applicable.

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
