# Peer review of "Bond Durability of Two-Step HEMA-Free Universal Adhesive"

_jfb, 2022, doi:10.3390/jfb13030134_

Round 1

Reviewer 1 Report

Dear authors,

Thanks for your submission. Adhesion to tooth structure is an important issue for dentists and researchers alike, and better understanding of different systems performances is needed for future development of the bonding agents. Kindly find a list of comments in the attached PDF document. Besides the comments in the attached file, find the following general comments:

1.  iThenticate shows that the similarity index of the manuscript is over 33%.

2. Consider minimizing the self-citation. 

Author Response

Thank you very much for giving your valuable comments and time to improve this paper. We have carefully considered to revise the paper and added our reply in attached PDF. 

Best regards,

Akimasa Tsujimoto

Reviewer 2 Report

The aim of the study “Bond Durability of 2-Step HEMA-free Universal Adhesive” was to compare shear bond strength and bond durability of a 2-step HEMA-free universal adhesive to 4 representative adhesives (etch and rinse, self-etch and universal with HEMA)

The study is well designed, results are well presented, and the results have clear clinical implication.

There are several issues that should be addressed:

Line 39 glycerol phosphate dimethacrylate (GPDM) is a hydrophilic monomer and is used in SE adhesives. Please check:

Wang R, Shi Y, Li T, Pan Y, Cui Y, Xia W. Adhesive interfacial characteristics and the related bonding performance of four self-etching adhesives with different functional monomers applied to dentin. J Dent. 2017 Jul;62:72-80. doi: 10.1016/j.jdent.2017.05.010. Epub 2017 May 17. PMID: 28527812.

Yoshihara K, Nagaoka N, Hayakawa S, Okihara T, Yoshida Y, Van Meerbeek B. Chemical interaction of glycero-phosphate dimethacrylate (GPDM) with hydroxyapatite and dentin. Dent Mater. 2018 Jul;34(7):1072-1081. doi: 10.1016/j.dental.2018.04.003. Epub 2018 Apr 30. PMID: 29716740.

Line 42 Are there any references to confirm the claims in this sentence “Despite significant developments in the field and the release of novel adhesive systems in the 25 years since OptiBond FL was released, no new adhesive has offered clearly superior bond performance, and so it is still the adhesive of choice in many hospital systems.”

Line 49 The sentence is not clear enough since 10-MDP forms chemical bonds with Ca and nano layers of 10-MDP-Ca were visualized using x-ray diffraction (Wang et al. 2017). Maybe it would be better to say “The primers and adhesives of these systems typically include 10-MDP which creates a strong and stable chemical bond with substrates’ calcium forming  nanolayers of 10-MDP-Ca.“

Line 80 Since this is in vitro study, maybe the P in PICO should be “problem“instead of “population“

Line 94 Please state that the universal adhesives were used in both modes after the sentence “Adhesives were studied in either etch-and-rinse or self-etch 93 mode according to the manufacturers’ instructions.“

Line 96 Add the information in Table 1 that Scotchbond Universal Plus Ad[1]hesiv is a 1 step Universal adhesive

Table 2 first row, last column “ratio“ instead of “ration“

Line 206 instead of “etch-and-rinse” there should be “self-etch”

Line 276-279 Elaborate which aspects and give references. “However, it is important to consider all aspects of adhesion if further improvements in adhesive dentistry are to be achieved. Indeed, compatible hydrophilicity between the adhesive agent, and the primed or non-primed etched enamel surface may be important“

Author Response

The aim of the study “Bond Durability of 2-Step HEMA-free Universal Adhesive” was to compare shear bond strength and bond durability of a 2-step HEMA-free universal adhesive to 4 representative adhesives (etch and rinse, self-etch and universal with HEMA)

The study is well designed, results are well presented, and the results have clear clinical implication.

RE: Thank you very much for your time and comments to improve this paper. 

There are several issues that should be addressed:

Line 39 glycerol phosphate dimethacrylate (GPDM) is a hydrophilic monomer and is used in SE adhesives. Please check:

RE: We have revised and added the paper as reference which you suggested below. 

Wang R, Shi Y, Li T, Pan Y, Cui Y, Xia W. Adhesive interfacial characteristics and the related bonding performance of four self-etching adhesives with different functional monomers applied to dentin. J Dent. 2017 Jul;62:72-80. doi: 10.1016/j.jdent.2017.05.010. Epub 2017 May 17. PMID: 28527812.

Yoshihara K, Nagaoka N, Hayakawa S, Okihara T, Yoshida Y, Van Meerbeek B. Chemical interaction of glycero-phosphate dimethacrylate (GPDM) with hydroxyapatite and dentin. Dent Mater. 2018 Jul;34(7):1072-1081. doi: 10.1016/j.dental.2018.04.003. Epub 2018 Apr 30. PMID: 29716740.

Line 42 Are there any references to confirm the claims in this sentence “Despite significant developments in the field and the release of novel adhesive systems in the 25 years since OptiBond FL was released, no new adhesive has offered clearly superior bond performance, and so it is still the adhesive of choice in many hospital systems.”

RE: We have added a reference.

Line 49 The sentence is not clear enough since 10-MDP forms chemical bonds with Ca and nano layers of 10-MDP-Ca were visualized using x-ray diffraction (Wang et al. 2017). Maybe it would be better to say “The primers and adhesives of these systems typically include 10-MDP which creates a strong and stable chemical bond with substrates’ calcium forming  nanolayers of 10-MDP-Ca.“

RE: We have revised the sentence and added the reference.

Line 80 Since this is in vitro study, maybe the P in PICO should be “problem“instead of “population“

RE: We have revised.

Line 94 Please state that the universal adhesives were used in both modes after the sentence “Adhesives were studied in either etch-and-rinse or self-etch 93 mode according to the manufacturers’ instructions.“

RE: We have added suggested sentence.

Line 96 Add the information in Table 1 that Scotchbond Universal Plus Ad[1]hesiv is a 1 step Universal adhesive

RE: We have revised. 

Table 2 first row, last column “ratio“ instead of “ration“

RE: We have revised.

Line 206 instead of “etch-and-rinse” there should be “self-etch”

RE: We have revised.

Line 276-279 Elaborate which aspects and give references. “However, it is important to consider all aspects of adhesion if further improvements in adhesive dentistry are to be achieved. Indeed, compatible hydrophilicity between the adhesive agent, and the primed or non-primed etched enamel surface may be important“

RE: We have added the reference.

Reviewer 3 Report

The work is definitely worth publishing, but the preparation of samples requires a better description in the context of the possible impact on the result. 

Please provide the shape and mounting scheme of samples cut from the tooth with dimensions. Report the area for calculating the averaged shear stress that you used as the shear bond strength in MPa  

What was the accuracy and dimensional deviation of the tooth specimens and their positioning in acrylic resin?  

Positioning in the sense of positioning and centering nito the ring with dimensions and accuracy, and positioning in relation to the directional properties of the tooth. Was it the direction controlled and repeatable or random? If random, what is the statistical influence on the result. 

 What modulus or at least type of acrylic resin for embedding tooth sample?  
The method of fixation/support the phenolitic ring and the dimension of the loading rod.  

This is necessary data for comparative purposes. I cannot find this data in [6], besides, the reader should not be referred to other works on fundamental methodology issues, especially if they are not open access.  

In the motivation of the research, please clearly indicate what is new compared to the work [6] (enamel/dentin ? SEM structure of bond zone)  
 And emphasize the new developments in the conclusions. In the discussion, it is worth emphasizing the limitations of the methodology, because the samples run dry or even in a liquid are not able to fully simulate the chemical aging phenomena occurring in the oral cavity in a shortened time.  

Whether the treatment of 4000 grid paper can have an effect on the actual roughness ?

Author Response

The work is definitely worth publishing, but the preparation of samples requires a better description in the context of the possible impact on the result. 

RE: Thanks you very much for your positive comments and time to improve the paper.

Please provide the shape and mounting scheme of samples cut from the tooth with dimensions. Report the area for calculating the averaged shear stress that you used as the shear bond strength in MPa  

RE: The detail of testing in figure was reported cited references of #19 and added normal shear bond strength to determine the initial load for fatigue test.

What was the accuracy and dimensional deviation of the tooth specimens and their positioning in acrylic resin? Positioning in the sense of positioning and centering nito the ring with dimensions and accuracy, and positioning in relation to the directional properties of the tooth. Was it the direction controlled and repeatable or random? If random, what is the statistical influence on the result. What modulus or at least type of acrylic resin for embedding tooth sample?  The method of fixation/support the phenolitic ring and the dimension of the loading rod.  

RE: We have added reference including illustration of the testing and indicated article including photo set-up of testing below. Dimension of phenolic ring is 25mm diameter and height. The information of acrylic resin was included in the paper.

Oper Dent 2015; 40(4): 379-395.

This is necessary data for comparative purposes. I cannot find this data in [6], besides, the reader should not be referred to other works on fundamental methodology issues, especially if they are not open access.  

RE: This cited reference is open access and free to access, so comparative data is available. 

In the motivation of the research, please clearly indicate what is new compared to the work [6] (enamel/dentin ? SEM structure of bond zone)  
 And emphasize the new developments in the conclusions. In the discussion, it is worth emphasizing the limitations of the methodology, because the samples run dry or even in a liquid are not able to fully simulate the chemical aging phenomena occurring in the oral cavity in a shortened time.

RE: We have added the sentence of limitation and motivation of the reserach in introduction and discussion sections.

Whether the treatment of 4000 grid paper can have an effect on the actual roughness ?

RE: In order to minimize the surface roughness of base surface, we used highly polished surfaces for the testing and make this clear in methodology section. 

Reviewer 4 Report

The manuscript titled “Bond Durability of 2-Step HEMA-free Universal Adhesive" presents a survey assessing bond durability in terms of fatigue bond strength  of a 2-step HEMA-free universal adhesive compared to representative adhesives in each systematic category.

The question is original and well defined; the results provide an advance in current knowledge; the results are interpreted appropriately, and they are significant; all conclusions are justified and supported by the results.

The article is written in an appropriate way; the data and analyses are presented appropriately.

The study is correctly designed and technically sound; the analyses are performed with the highest technical standards; the data are robust enough to draw the conclusions; the methods, tools, software, and reagents are described with sufficient details to allow another researcher to reproduce the results.

The conclusions are interesting for the readership of the Journal; the paper will attract a wide readership.

There is an overall benefit to publishing this work; the work provides an advance towards the current knowledge; the authors addressed an important long-standing question with smart experiments.

The English language is appropriate and understandable.

I would make the following recommendations to the authors:

Please specify what HEMA stand for.

Please specify which university the Institutional Review Board that approved the study belongs to

Figures 1, 2, 7, and 8 are of very low quality.

Therefore, my recommendation is accept after minor revision.

Author Response

Please specify what HEMA stand for.

RE: We have revised.

Please specify which university the Institutional Review Board that approved the study belongs to

RE: We have added sentences.

Figures 1, 2, 7, and 8 are of very low quality.

RE: We have changed figures which have higher resolution.

Therefore, my recommendation is accept after minor revision.

RE: Thank you very much for your positive review.

Round 2

Reviewer 3 Report

Dear Authors, 
You did not explain the basic simple questions at all, and you refer me to other Refs where it is not easy to find.  
I don't need to see the test device which is in [1] but there are no samples and its dimensions.  
A diameter of 25 mm is an awful lot, and I was surprised how you locate the tooth sample.  
Finally found the dimensions in 

Effect of Phosphoric Acid Pre-etching on Fatigue Limits of Self-etching Adhesives | Operative Dentistry (allenpress.com) 
“ Metal rings machined from 304 stainless steel with an inner diameter of 2.4 mm, an outer diameter of 4.8 mm, and a length of 2.6 mm were used to bond resin composite (Z100 Restorative) to enamel/dentin surfaces for SBS and SFL tests. The bonding procedure resulted in a resin composite cylinder inside the ring that was 2.36 mm in diameter and approximately 2.5 mm in length. The ring was left in place for the tests.” 

but that still doesn't explain.  

Is it so difficult to include a simple sample diagram with dimensions?  
Why should the reader search for basic information?  
I understand that I give Refs to some extra info but to basic?  
What's worse, the comma moved from 2.5mm to 25mm? 
Therefore, I did not understand where the tooth tissue was in it and what its area is in the F/A formula.  
I don't want to worry anymore about how the dimensional error of the tooth cut (real area A mm2) affects the result in MPa.  

The direction of the dentin and tubular arrangement is not clear. The figs clearly show that the arrangement of the tubes is random.  
So the question is, do you have REFs which prove it has no effect? If it has what? If you do not have this data in your research, at least you should describe it and state it as a limitation of the research methodology.  

In addition, I see 200Hz in the new version. How do you know that 200Hz can be used and whether it will not start the thermal effect, which may be different in relation to different materials, and it does not exist in the oral cavity.

I hope you will complete it, because otherwise it's a good job, but you cannot ignore the missing information and variables that may affect the result, especially selectively with regard to the tested materials

Author Response

Dear Authors, 
You did not explain the basic simple questions at all, and you refer me to other Refs where it is not easy to find.  
I don't need to see the test device which is in [1] but there are no samples and its dimensions.  
A diameter of 25 mm is an awful lot, and I was surprised how you locate the tooth sample.  
Finally found the dimensions in 

Effect of Phosphoric Acid Pre-etching on Fatigue Limits of Self-etching Adhesives | Operative Dentistry (allenpress.com) 
“ Metal rings machined from 304 stainless steel with an inner diameter of 2.4 mm, an outer diameter of 4.8 mm, and a length of 2.6 mm were used to bond resin composite (Z100 Restorative) to enamel/dentin surfaces for SBS and SFL tests. The bonding procedure resulted in a resin composite cylinder inside the ring that was 2.36 mm in diameter and approximately 2.5 mm in length. The ring was left in place for the tests.” but that still doesn't explain.  

Is it so difficult to include a simple sample diagram with dimensions?

 Why should the reader search for basic information?  
I understand that I give Refs to some extra info but to basic?  
What's worse, the comma moved from 2.5mm to 25mm? 
Therefore, I did not understand where the tooth tissue was in it and what its area is in the F/A formula.  
I don't want to worry anymore about how the dimensional error of the tooth cut (real area A mm2) affects the result in MPa.  

RE: We address this set of comment below in the follow-up that the reviewer sent. We apolgoize on our part for any confusion. We are not sure about the 25mm comment but affirm that the phenolic rings with acrylic resin the teeth are placed in are 25mm.

The direction of the dentin and tubular arrangement is not clear. The figs clearly show that the arrangement of the tubes is random.  
So the question is, do you have REFs which prove it has no effect? If it has what? If you do not have this data in your research, at least you should describe it and state it as a limitation of the research methodology.  

RE: We address this limitation below in the follow-up that the reviewer sent.

In addition, I see 200Hz in the new version. How do you know that 200Hz can be used and whether it will not start the thermal effect, which may be different in relation to different materials, and it does not exist in the oral cavity.

RE: The reviewer sent additional information and stated 20 Hz is ok. This is in agreement with our previous work (cited below) that shows that 20 Hz, which is accelerated, provides similar results as 2 Hz, which is physiological.

Tsujimoto A, Barkmeier WW, Erickson RL, Fischer NG, Markham MD, Takamizawa T, Latta MA, Miyazaki M. Shear fatigue strength of resin composite bonded to dentin at physiological frequency. Eur J Oral Sci. 2018 Aug;126(4):316-325.

I hope you will complete it, because otherwise it's a good job, but you cannot ignore the missing information and variables that may affect the result, especially selectively with regard to the tested materials

RE: We appreciate the time and effort the reviewer put into our manuscript to make it clearer.

Further comments From Reviewer 3:

Please send to Authors that I saw it wrong Hz

and 20Hz is ok !

And I found the diagram in [19],

but I think they should include one anyway, because it explains a lot in a moment.

However, the repeatability with respect to the dentine direction is not known.

RE: We have addressed the 2 vs. 20 Hz. question above and appreciate the clarification from the reviewer.

Based on this comment we have added a version of the diagram from [19] to this manuscript as Figure 1. The actual surface area is 4.37 mm2  that is used to calculate the mPa in the F/A equation. We added this to the manuscript. We are also including an actual image of the setup below. The tooth is placed in phenolic rings that are 25 mm in diameter with the tooth in the middle. This helps us polish the sample and make them easy to handle and manipulate. Actual image of the experimental for the shear fatigue bond strength testing was attached in PDF.

Finally, we attempt to place the teeth in the acrylic resin (Fastry Custom Tray and Acrylic Base Plate Material) so that the dentinal tubules are perpendicular to the load being applied. We polish the facial or lingual side and then mount the tooth to apply the load from coronal to apical. In other words, we try to polish so that the tubules are vertical when we apply the adhesive on the benchtop. We added two citations to the paper about this that state this orientation results in the largest shear bond strength values. We do this for better comparison between adhesives. However, we admit we cannot precisely control this so we have have added this as a limitation of the study in the discussion. We thank the reviewer for pointing this out.
